# Neuroprotection and Disease Modification by Astrocytes and Microglia in Parkinson Disease

**DOI:** 10.3390/antiox11010170

**Published:** 2022-01-17

**Authors:** Shinichi Takahashi, Kyoko Mashima

**Affiliations:** 1Department of Neurology and Stroke, Saitama Medical University International Medical Center, 1397-1 Yamane, Hidaka-shi 350-1298, Japan; 2Department of Physiology, Keio University School of Medicine, 35 Shinanomachi, Shinjuku-ku, Tokyo 160-8582, Japan; kymashima@gmail.com; 3Department of Neurology, Tokyo Saiseikai Central Hospital, 1-4-17 Mita, Minato-ku, Tokyo 108-0073, Japan

**Keywords:** astrocyte, astroglia, glutathione, lactate, microglia, Toll-like receptor 4

## Abstract

Oxidative stress and neuroinflammation are common bases for disease onset and progression in many neurodegenerative diseases. In Parkinson disease, which is characterized by the degeneration of dopaminergic neurons resulting in dopamine depletion, the pathogenesis differs between hereditary and solitary disease forms and is often unclear. In addition to the pathogenicity of alpha-synuclein as a pathological disease marker, the involvement of dopamine itself and its interactions with glial cells (astrocyte or microglia) have attracted attention. Pacemaking activity, which is a hallmark of dopaminergic neurons, is essential for the homeostatic maintenance of adequate dopamine concentrations in the synaptic cleft, but it imposes a burden on mitochondrial oxidative glucose metabolism, leading to reactive oxygen species production. Astrocytes provide endogenous neuroprotection to the brain by producing and releasing antioxidants in response to oxidative stress. Additionally, the protective function of astrocytes can be modified by microglia. Some types of microglia themselves are thought to exacerbate Parkinson disease by releasing pro-inflammatory factors (M1 microglia). Although these inflammatory microglia may further trigger the inflammatory conversion of astrocytes, microglia may induce astrocytic neuroprotective effects (A2 astrocytes) simultaneously. Interestingly, both astrocytes and microglia express dopamine receptors, which are upregulated in the presence of neuroinflammation. The anti-inflammatory effects of dopamine receptor stimulation are also attracting attention because the functions of astrocytes and microglia are greatly affected by both dopamine depletion and therapeutic dopamine replacement in Parkinson disease. In this review article, we will focus on the antioxidative and anti-inflammatory effects of astrocytes and their synergism with microglia and dopamine.

## 1. Introduction

The pathophysiology and pathogenesis of Parkinson disease involves the depletion of dopamine following the degeneration and loss of dopaminergic neurons by undetermined causes in the substantia nigra [1,2,3]. In addition, loss of dopaminergic neurons in the ventral tegmental area (VTA) of the midbrain, which form mesolimbic and mesocortical projections, causes non-motor symptoms, such as emotional and cognitive impairment. Interestingly, however, the degenerative processes in the VTA are milder and slower, suggestive of different mechanisms operating [4,5,6,7]. Mitochondrial dysfunction [8,9] and alpha-synuclein deposition [10,11] are thought to play pivotal roles. In the striatum, which is the main projection site of these dopaminergic neurons, released dopamine is maintained at appropriate concentrations. At the basal level, dopamine release is maintained at a constant level by tonic release; however, when necessary, dopamine is rapidly released (phasic release) to regulate motor function appropriately. In fact, substantia nigra dopaminergic neurons exhibit a 2–4 Hz pacemaking activity at the basal level [12,13,14]. The neuronal activity reflects the action potential (the rapid influx of Na^+^ into the neuron). As a preparatory step, neurons must maintain an inward Na^+^ concentration gradient from extracellular to intracellular. The mitochondria, which are the primary sites of adenosine triphosphate (ATP) production, are overloaded by this neuronal activity and are prone to generate reactive oxygen species (ROS). In brain regions that require continuous neuronal activity, including the pacemaking activity of dopaminergic neurons, the ROS generated in neurons can cause secondary damage to their own mitochondria, resulting in local dysfunction and regional neurological symptoms [13,14]. The possibility that such excessive neuronal activity may trigger the degeneration of various neurons has long been recognized, and hyperexcitability may contribute to disease exacerbation by inducing the degeneration of both motor neurons and dopaminergic neurons. In amyotrophic lateral sclerosis (ALS), a neurodegenerative disease in which the motor neurons are the main loci of disease activity, the suppression of hyperexcitability has been investigated as a therapeutic strategy [15,16,17]. Clinical trials using antiepileptic drugs has already been completed [18,19,20,21,22]. In Parkinson disease, the possibility of disease-modifying therapy using Ca^2+^ channel blockers that inhibit neuronal excitability has been suggested [23,24,25]. Moreover, milder and slower neurodegenerative processes of dopaminergic neurons in the VTA may reflect less Ca^2+^-related oxidative stress as compared with those in the substantia nigra [4,5,6,7]. The distribution of amyloid-beta accumulation in the brain, which underlies the pathogenesis of Alzheimer disease, has also been linked to neural activity [26,27]. The medial frontal lobe, posterior cingulate gyrus, and precuneus, which are the most common sites of amyloid-beta accumulation, are reportedly associated with the default mode network, i.e., sites where brain activity is maintained in the basal state [26,27].

Neuronal hyperexcitability promotes the release of neurotransmitters from presynaptic neurons. In Parkinson disease, the released dopamine itself may be chemically responsible for inducing oxidative stress and predisposing the brain to neurodegeneration [13,14]. In contrast, glutamate is widely distributed in the brain and acts as an excitatory neurotransmitter in ALS [15,16,17,18,19,20,21,22]. Glutamate acts on N-methyl-D-aspartate (NMDA) receptors at postsynaptic sites, but excitotoxicity, caused by excessive glutamate release, leads to neuronal death. This mechanism is also thought to be the core mechanism responsible for neuronal death associated with ischemic stroke [28,29,30,31]. The same excitotoxicity can also be induced in lower motor neurons bearing glutamate receptors, since upper motor neurons in the motor cortex utilize glutamate as a neurotransmitter. The overexcitation of upper motor neurons in the cortical motor cortex causes oxidative stress in their mitochondria; the released glutamate then causes excitatory cell death in lower motor neurons in the anterior horn of the spinal cord [15,16,17]. However, glutamate released into the synaptic cleft is immediately recovered by glutamate transporters expressed in the end-feet of astrocytes surrounding the synapse, and the elevated extracellular concentration is transient, meaning that postsynaptic neuronal damage is unlikely to occur during normal neural activity [28,29,30,31]. In this situation, metabolic mechanisms within astrocytes process glutamate metabolism rapidly. In contrast, dopamine is re-taken up by the dopaminergic neurons themselves and has little metabolic interaction with astrocytes [32,33]. This difference is thought to be relevant to the pathogenesis of Parkinson disease, especially in association with alpha-synuclein toxicity [13,14,29,30,31,32,33,34,35].

## 2. Dopaminergic Neurons and Astrocytes

After release in the striatum, dopamine acts on dopamine receptors expressed on medium spiny neurons [36,37]. Here, the dopamine D2 receptor (D2R) is associated with Gi protein, and its action is inhibitory. The dopamine D3, 4 receptors, which are expressed at low levels in mesolimbic and mesocortical systems, are also inhibitory [4,5,6,7]. How the expression of D3,4 receptors in the VTA dopaminergic system affect the pathological processes remains to be determined. Therefore, dopamine-releasing neurons are unlikely to cause excitotoxicity in medium spiny neurons [36,37]. In addition, nigrostriatal dopaminergic neurons themselves express D2R, and the released dopamine is thought to exert autocrine inhibitory effects on the excitability of these neurons [38,39,40]. Dopaminergic neurons perform the tonic release of dopamine according to the pacemaking activity and switch to phasic release when necessary; the released dopamine may exert negative feedback on dopaminergic neurons. This can be interpreted as an endogenous protective mechanism of dopamine to suppress the overexcitation of dopaminergic neurons. However, compensation for the reduction in dopamine release is needed in Parkinson disease to mask the development of motor dysfunction by reducing excitatory inhibition. These compensations ultimately increase neuronal excitation, hastening cell death. The expression of D2R on astrocytes and microglia is discussed below.

Of note is that dopamine, as a neurotransmitter, per se, causes oxidative stress [32,33], but glutamate does not [29,33]. Dopamine undergoes autoxidation and is converted to dopamine quinone, an oxidative stress-causing substance that binds to various proteins and leads to their dysfunction [29,32,33]. Unlike glutamate, dopamine neurons themselves are responsible for the re-uptake of dopamine after cellular release [41,42,43,44]. Na^+^-dependent monoamine transporters, such as dopamine transporter (DAT), norepinephrine transporter (NET), and serotonin transporter (SERT), are expressed in nerve endings and are involved in dopamine retrieval [43,44]. A DaTscan, which is a clinically applicable functional radiological imaging technique used in patients suspected of having Parkinson disease, visualizes decreases in dopamine reuptake caused by the degeneration of dopaminergic neurons [45,46]. Although damage to dopaminergic neurons eventually results in cell death, the initial stages of degeneration occur in the nerve endings [47], making a DaTscan of high diagnostic value in clinical settings.

Astrocytes form a tripartite synapse with pre- and post-synaptic neurons, and the glutamatergic synapse is enveloped by astrocyte end-foot processes [29,30,31]. The anatomical location of astrocytes in the striatum in relation to the terminal portions of dopamine neurons and the receptor portions of medium spiny neurons has not been adequately studied [48]. The dopamine transporter expressed in striatal astrocytes, however, has been studied extensively [43,44]. Based on immunohistological studies involving measurements of protein expression (Western blotting) and mRNA (RT-PCR), these multiple monoamine transporters are thought to be expressed in striatal astrocytes and to play certain functional roles [29,33,43,44].

## 3. Effects of Glutamate and Dopamine on Astrocytes

With respect to glutamate, re-uptake after neuronal release into the synaptic cleft is performed by astrocytes, but not by neurons (Figure 1). This is of great importance for the coupling of astrocyte and neuronal energy metabolism and the role of astrocytes in reducing mitochondrial oxidative stress. The glutamate transporter in astrocytes takes up glutamate in a Na^+^-dependent manner using an inward concentration gradient into the cell. After glutamate uptake, Na^+^,K^+^-ATPase, which is activated to restore the Na^+^ concentration gradient, consumes ATP, thereby enhancing glucose utilization in astrocytes at the expense of glutamate recovery. Astrocyte foot processes surround synapses on one side (tripartite synapse) and coat the surface of brain capillaries on the other. This anatomical arrangement is optimal for the efficient uptake of intravascular glucose and its use as an energy substrate for ATP production. Synapses, astrocytes, and brain microvessels form a single functional unit (i.e., a neurovascular unit) [29,30,31]. Although the function of the neurovascular unit in general has been studied extensively, the function of the neurovascular unit composed of dopaminergic neurons is not well known [29,33,48]. The pathway for ATP production from glucose in astrocytes during glutamate uptake consists of a glycolytic system in the cytosol, followed by oxidative phosphorylation in mitochondria [29,30,31]. The involvement of astrocytes in neuronal energy metabolism has attracted much attention since it was first proposed by Pellerin and Magistretti in 1994 [49], and has been confirmed by several researchers [50,51,52,53]. However, its physiological significance remains controversial, and some reports are supportive [50,51,52,53,54,55,56,57,58,59,60], while others are not [61,62,63,64,65,66,67,68,69,70,71]. The uptake of glutamate by astrocytes after its release in response to neuronal excitation acts as a metabolic signal to astrocytes, enhancing glucose utilization and simultaneously activating both the glycolytic system and mitochondria [29,30,31]. Based on in vitro and in vivo data, the activation of the glycolytic system is dominant in astrocytes, and the lactate produced in astrocytes is supplied to neurons, which produce ATP as an energy substrate; this concept is known as the astrocyte–neuron lactate shuttle (ANLS) model [49,50,51,52,53,54,55,56,57,58,59,60,61,62,63,64,65,66,67,68,69,70,71] (Figure 1).

Two things are important in the ANLS model. First, the increase in astrocyte glycolysis associated with the recovery of released glutamate leads to a concomitant increase in flux to the pentose–phosphate pathway (PPP), a shunt pathway of the glycolytic system [29,30,31] (Figure 1). The PPP increases the ratio of intracellular reducing equivalents of NADPH, which are essential for the maintenance of the reduced form of glutathione (GSH), an antioxidative compound in the brain. In other words, the activation of the glycolytic system and PPP flux together act as a brain protective function to reduce oxidative stress. Second, lactate released from astrocytes upon glycolytic-dominant activation is taken up by neurons via the monocarboxylate transporter (MCT) and converted to pyruvate. After conversion to pyruvate, lactate is metabolized oxidatively in the mitochondria to produce ATP [29,30,31] (Figure 1). Simultaneously, lactate inhibits neuronal activity via the hydroxycarboxylic acid receptor 1 (HCAR1), expressed in neurons [29,30,31,72,73,74]. Whether the ANLS model can be adapted to understand the activities of dopaminergic neurons in healthy people and patients with Parkinson disease is an important question.

Glutamate (Glu) is a ubiquitous excitatory neurotransmitter that causes cell death through the N-methyl-D-aspartate (NMDA) receptor-mediated increases in intracellular Ca^2+^ concentration ([Ca^2+^]_i_), activation of neuronal nitric oxide synthase (nNOS), and the production of nitric oxide (NO). Astrocytes take up glutamate by Na^+^-dependent glutamate transporters, which stimulates astrocytic Na^+^,K^+^-ATPase, thereby enhancing glycolytic metabolism (aerobic glycolysis). The lactate/pyruvate fuels the neuronal tricarboxylic acid (TCA) cycle (astrocyte–neuron lactate shuttle model). Mitochondria are a major source of reactive oxygen species (ROS), which injure neurons. Glutamate in astrocytes is converted into glutamine (Gln) and then recycled back to neurons (glutamate–glutamine cycle). Glu-derived α-ketoglutarate (αKG) serve as an energy substrate. The activation of glycolysis in astrocytes increases flux into the pentose–phosphate pathway (PPP), a shunt pathway of glycolysis, at glucose 6-phosphate (G6P) in astrocytes. The transcriptional regulation of the rate-limiting enzyme of PPP, G6P dehydrogenase, is under the Kelch-like enoyl-CoA hydratase-associated protein 1 (Keap1)/nuclear factor erythroid 2 p45 subunit-related factor 2 (Nrf2) system. The activation of PPP increases the ratio of the reduced form and the oxidized form of nicotinamide adenine dinucleotide phosphate (NADPH/NADP^+^), which is used to convert the oxidized form of glutathione (GSSG) to the reduced form of glutathione (GSH). GSH can be transferred to neurons from astrocytes as amino acid components of GSH; i.e., glutathione Gln, cysteine (Cys), and glycine (Gly).

## 4. Dopamine and Astrocyte Activation of the Glycolytic System and PPP

As mentioned above, astrocytes can express three types of transporters (DAT, NET, and SERT) capable of transporting dopamine in a Na^+^-dependent manner [43,44] (Figure 2). We examined glucose utilization in rat cultured astroglia (astroglia prepared from the striatum and cerebral cortex) in vitro before and after dopamine application using the [^14^C]deoxyglucose technique. Lactate release in the cell culture medium, which is an index of the utilization of the glycolytic system over oxidative glucose metabolism in mitochondria, was simultaneously measured. Within one hour after dopamine administration, glucose utilization in the cultured astroglia increased rapidly and was accompanied by an increase in lactate production [29,33]. These responses were thought to reflect the increased utilization of the glycolytic system in astroglia (Figure 2). Similarly, glucose utilization was also increased in cultured rat striatal neurons, indicating that ATP consumption is closely associated with dopamine reuptake by neurons [29,31]. The magnitude of the increase in glucose consumption in neurons was lower than that in astroglia, suggesting that the pathway by which ATP is produced in neurons depends mainly on mitochondrial metabolism (Figure 2), which is more efficient at producing ATP than the glycolytic system and requires less glucose [29,31]. Whether lactate is preferred over glucose as a tricarboxylic acid (TCA) cycle energy substrate in dopaminergic neurons remains unconfirmed (Figure 2).

Recently, Hayakawa et al. reported observing the phenomenon of mitochondria being donated from astrocytes to neurons [75,76,77,78,79,80,81,82]. We have discussed the transfer of metabolites from astrocytes to neurons [13,29,31], which demonstrates how astrocytes create an efficient and viable environment in the metabolic system of neurons, whose primary goal is efficient production of ATP. In fact, mitochondria are constantly exposed to oxidative stress in the process of oxidative metabolism of glucose. Oxidative stress is one of the most important factors implicated in the pathogenesis of many neurodegenerative diseases for which age-related mitochondrial dysfunction is the most important risk factor. The mechanism by which mitochondria are supplied from astrocytes to neurons is considered to represent the strongest compartmentalization between astrocytes and neurons. At present, it has been confirmed that mitochondria are activated after ischemic stroke, but the universality of this mechanism in degenerative diseases still needs to be examined.

Ca^2+^ and ATP levels are the main triggers for mitochondrial biogenesis in neural cell body, in a mechanism dependent on the peroxisome proliferator-activated γ co-activator-1α (PGC-1α)-nuclear respiration factors 1 and 2 (NRF-1 and NRF-2)-mitochondrial transcription factor A (TFAM) pathway. PGC-1α is the master regulator of mitochondrial biogenesis. It activates NRF-1 and NRF-2, leading to the expression of several mitochondrial genes, including proteins that are required for mtDNA transcription and replication, namely TFAM [83,84]. Additionally, in astrocytes, mitochondriogenesis seems to be regulated by the similar mechanisms [83,84]. Interestingly, amyloid-beta_1–42_, which induces apoptosis in neurons but not in astrocytes. In astrocytes amyloid-beta_1–42_ decreases protein expressions of sirtuin 1 (SIRT-1) and peroxisome proliferator-activated receptor γ (PPAR-γ) and over-expresses PGC-1α and TFAM, protecting mitochondria against amyloid-beta_1–42_-induced damage and promoting mitochondrial biogenesis. The effects of alpha-synuclein on astrocytic mitochondriogenesis seem to be worth examining. The activation of biogenesis and transfer of astrocytic mitochondria to neurons may lead to a promising therapeutic strategy for neurodegenerative diseases.

We previously mentioned the autoxidation of dopamine after neuronal release and the oxidative stress resulting from this product, dopamine quinone [32]. The reduced form of glutathione (GSH) plays an important role as an antioxidant. GSH exerts its antioxidant effect by converting itself to the oxidized form of glutathione (GSSG), which is then converted back to GSH by nicotinamide adenine dinucleotide phosphate (NADPH), enabling GSH to re-exert its antioxidant effect. Since NADP^+^, which is produced from NADPH when GSSG is converted to 2 GSH, is reduced back to NADPH by the action of PPP, the high PPP flux in astrocytes is a sign of the high antioxidant activity of astrocytes [29,30,31]. Astrocytes can be said to play a role in protecting neurons against oxidative stress through metabolic coupling. In addition, astrocytes are also responsible for exerting antioxidant activity in the extracellular space by releasing GSH [29,32,33]. In fact, one of the most important roles of GSH is in the reduction of cystine to cysteine, which is used for the synthesis of glutathione in neurons [29,32,33].

In the striatum, the optimal dopamine concentration is maintained by the pacemaking activity of dopaminergic neurons. However, this constant activity can elicit mitochondrial overload, leading to the increased production of reactive oxygen species (ROSs). The reduced form of glutathione (GSH) plays an important role in eliminating ROSs. The synthesis of glutathione requires glutamine (Gln)/glutamate (Glu), which is supplied by astrocytes, as well as cysteine (Cys) and glycine (Gly), which are also derived from GSH released from astrocytes. Thus, neuronal GSH synthesis is largely dependent on astrocytes. Most of released dopamine is taken up by Na^+^-dependent dopamine transporter (DAT) expressed in presynaptic dopaminergic neurons. However, the astrocytes that surround the synapses also seem to take up dopamine via norepinephrine transporter (NET) and serotonin transporter (SERT). Increases in the intracellular Na^+^ concentration ([Na^+^]_i_) activate Na^+^,K^+^-ATPase, thereby enhancing glycolytic metabolism (aerobic glycolysis) in astrocytes. In astrocytes, the pentose–phosphate pathway (PPP), a shunt pathway of glycolysis, is co-activated; this process facilitates the conversion of NADP^+^ to NADPH. NADPH is necessary for glutathione reductase to eliminate ROSs, since glutathione peroxidase requires the reduced form of GSH converted from the oxidized form of glutathione (GSSG), which is dependent on NADPH. GSH released from astrocytes also plays an important role in erasing dopamine-derived ROSs in the extracellular space. A rate-limiting enzyme of the PPP, glucose 6-phosphate (G6P) dehydrogenase, is regulated transcriptionally by the Kelch-like enoyl-CoA hydratase-associated protein 1 (Keap1)/nuclear factor erythroid 2 p45 subunit-related factor 2 (Nrf2) system. Therefore, the pharmacological activation of the Keap1/Nrf2 system is expected to enhance astrocytic protective mechanisms against ROSs, leading to a novel therapeutic strategy for the treatment of Parkinson disease.

## 5. Glutathione Synthesis System and Dopaminergic Neurons

Glutathione is composed of three types of amino acids (glutamate, cysteine, and glycine). The glutathione synthesis capacity is known to be much higher in astrocytes than in neurons [29,32,33]. Among the three amino acids composing glutathione, glutamate is released as a neurotransmitter from glutamatergic neurons and is collected in astrocytes, where it is added to ammonia and synthesized into glutamine. Glutamine is then recycled back to the glutamatergic neurons, where it is degraded by glutaminase in presynaptic neurons and utilized as glutamate for neurotransmission (glutamate–glutamine cycle) [29,30,31]. Of note, glutamine can also be used for glutathione synthesis in neurons. Cysteine, which is also required for glutathione synthesis, exists as the autoxidized form cystine outside the cell. Astrocytes can directly take up cystine using the cystine/glutamate antiporter (xCT). In astrocytes, abundant cystine is immediately reduced to cysteine using reducing equivalents and GSH; glutamate and glycine are then added to complete GSH synthesis [32].

On the other hand, neurons do not express xCT, and cystine cannot be used for the synthesis of glutathione [32]. For this reason, neurons take up extracellularly reduced cysteine, rather than cystine. In dopaminergic neurons, most of the dopamine released as a neurotransmitter is reincorporated into neurons. Some dopamine may be taken up by astrocytes; unlike the situation for glutamate, however, astrocytes are not involved in glutathione synthesis, and no metabolic coupling occurs between them and neurons [29,33]. This means that astrocytes are less likely to provide a benefit to dopaminergic neurons. In Parkinson disease, after the progressive loss of dopaminergic neurons, the nerve endings are unable to retain dopamine, so some of the therapeutically administered dopamine precursor l-3,4-dihydroxyphenylalanine (l-DOPA) is converted to dopamine in astrocytes, where it is retained and released. However, the effect of maintaining a constant concentration of dopamine in the striatum eventually becomes insufficient, resulting in the on–off phenomenon and dyskinesia associated with concentration changes during l-DOPA administration, leading to insufficient improvements in activities of daily living [1,2,3].

## 6. PPP and Keap1/Nrf2 System

Although the importance of glutathione and PPP in maintaining the reduced state of astrocytes has been described, the flux to PPP, a shunt pathway of the glycolytic system, is not regulated solely by glucose utilization [85,86]. The enzyme that determines shunting to the PPP, glucose 6-phosphate dehydrogenase (G6PD), is regulated by transcription factors as well as allosteric regulation by upstream glycolytic glucose metabolism [13,29,30,31,33]. In addition to G6PD, the Kelch-like enoyl-CoA hydratase-associated protein 1 (Keap1)/nuclear factor erythroid 2 p45 subunit-related factor 2 (Nrf2) system is also involved in regulating the transcription of other enzymes, such as glutathione synthase and transferase [13,29,30,31,33]. The transcription factor Nrf2 binds to Keap1 in the cytoplasm, and this complex is ubiquitinated and always degraded by the proteasome system, so it does not exert transcriptional activity. When ROS, representing oxidative stress, is generated, ROS binds to the cysteine residue of Keap1 and changes its conformation so that it cannot maintain its binding state with Nrf2. Free Nrf2 is translocated from the cytoplasm to the nucleus, where it binds to an antioxidant response element (ARE) upstream of the gene being regulated. The subsequent transcription and translation of the enzyme initiates an antioxidant stress response [13,29,30,31,33].

We have confirmed that various Nrf2 activators promote the transcription of Keap1/Nrf2 system-dependent enzymes, such as G6PD in cultured astroglia; this initiates an increase in PPP flux within a very short time [33]. An increase in GSH is simultaneously induced in astroglia, suggesting that the state of the Keap1/Nrf2 system is an indicator of astrocyte antioxidant activity [33]. When cultured neurons and astroglia are compared, PPP activity is 5–7 times higher in astroglia than in neurons [87]. The antioxidant effect of astrocytes in the brain may be a complementary function for the elimination of ROS in neurons, which inevitably occurs because of their specialization in neural activity [13,29,30,31,33,85,86].

## 7. Parkinson Disease and the Keap1/Nrf2 System

Nrf2 activators are expected to inhibit the progression of Parkinson disease as well as other neurodegenerative diseases, and many basic studies have been conducted [88,89,90]. For example, in a mouse model of Parkinson disease, disease progression was accelerated in Nrf2 knockout mice, compared with wild-type mice [90]. Nrf2 activators such as sulforaphane, a naturally occurring plant compound, induce Nrf2 release by altering the conformation of Keap1 [33,48]. Bardoxolone methyl, a synthetic triterpenoid synthesized from oleic acid, has been developed as a candidate drug for the treatment of diabetic kidney disease and is expected to have Nrf2-activating activity [91,92]. These compounds cannot pass through the blood–brain barrier, making it difficult for them to exert their effects on central nervous system diseases. In contrast, ROS generated from neurons in the brain is an important endogenous signal that activates Keap1/Nrf2 in astrocytes. As mentioned above, dopamine itself induces oxidative stress and is expected to have a similar effect [29,32,33]. We loaded dopamine into cultured astroglia and measured PPP flux after 12 h [33]. Apart from the acute PPP flux increase associated with increased glucose consumption in response to dopamine uptake, an Nrf2-dependent increase was also observed. This response was assumed to result from the activation of the Keap1/Nrf2 system via dopamine receptors, although the generation of dopamine quinone by autoxidation acts on Keap1 in a manner similar to that of ROS [33].

The addition of dopamine to cultured striatal neurons induces cytotoxicity, which is not observed in astrocytes [33]. This difference is thought to be due to oxidative stress derived from dopamine quinone generated from dopamine, which can be suppressed by a combination of antioxidant drugs. In fact, no cytotoxicity was observed in striatal neurons when the same concentration of dopamine was added to co-cultures with astroglia [33]. These findings indicate that astroglia do, indeed, exert neuroprotective effects. Even when cultured astroglia prepared from Nrf2-knockout mice were co-cultured with wild-type mouse striatal neurons, the cytotoxicity in the striatal neurons was unaltered, strongly suggesting the involvement of Keap1/Nrf2 in the neuroprotective effect of astroglia [33]. Of course, ROS production upon dopamine addition occurs not only as a result of the autoxidation of dopamine itself, but also within dopaminergic neurons that recapture dopamine. It is not difficult to imagine that mitochondria-derived ROS production also occurs here because, unlike astrocytes, glucose metabolism in mitochondria is the main source of ATP production, which is required to restore the Na^+^ concentration gradient necessary for dopamine reuptake. Whether the antioxidant system in dopaminergic neurons is more vulnerable than in other neurons remains uncertain. However, the glutathione metabolic system is not coupled to the astrocytes that operate in dopaminergic neurons [29,30,31].

## 8. Injurious Astrocytes in Parkinson Disease

While astrocytes possess a protective effect on neurons through their antioxidant activity [29], they can also act in an injurious manner [93]. Damage-associated molecular patterns (DAMPs) are ligand stimuli that induce damaging astrocytes; DAMPs themselves are assumed to be products released from damaged neurons, such as mitochondrial fragments and organelles [94]. Recently, alpha-synuclein, a disease-specific marker of Parkinson disease, has been included among known DAMPs [11]. Toll-like receptor 4 (TLR4), which is widely expressed in mammalian nervous system cells including neurons, astrocytes, and microglia, is a cell surface receptor for DAMPs [94,95,96]. Nitric oxide (NO), together with ROS, is a gaseous mediator with strong cytotoxic effects. In ischemic neuronal injury, it is an important mediator of actual neuronal death downstream of glutamate toxicity [97,98]. Studies in rodents and gyrencephalic brain primate models have shown that blocking the signaling of NMDA receptors, postsynaptic density protein 95 (PSD-95), and NO synthase (NOS) has neuroprotective effects during the acute phase of cerebral infarction [97,98]. These effects have also been reported in human stroke patients. Recently, clinical trials in humans with acute cerebral infarction have suggested the clinical efficacy of NA-1 (nerinetide) that blocks the NMDA receptor signal to NOS activation [99,100,101].

Several experiments have shown that microglia and astrocytes treated with lipopolysaccharide (LPS), a classical ligand for TLR4, develop an inflammatory profile [94,95,96]. This is especially true for M1 microglia and A1 astrocytes. M1 microglia release several pro-inflammatory cytokines, which further enhance the inflammatory profile of astrocytes [102,103]. Therefore, the induction of injurious astrocytes via microglia is thought to be important in neuronal injury. We hypothesized that the astrocytic antioxidative effect (ROS scavenging mechanism through PPP) was responsible for the absence of the expected ROS generation in LPS-treated astroglial cultures [94]. In fact, PPP flux was increased in astroglia after LPS treatment, and the intracellular concentration of the reduced form of glutathione was elevated. The mRNA level of the gene heme oxygenase 1 (HO-1), which is an indicator of the activation of the Keap1/Nrf2 system, was also increased, suggesting the involvement of the Keap1/Nrf2 system in the expression of TLR4-induced antioxidant activity in astroglia [29,94].

## 9. Neuroinflammatory Microglia and Neuroprotective Astrocytes

There are two major mechanisms by which Nrf2 dissociates from Keap1 and exerts transcriptional activity: a conformational change in Keap1, an adaptor protein, and the phosphorylation of serine resides of Nrf2 [13,29]. With the former mechanism, the binding of ROS to cysteine residues is important (Figure 3), and the ROS generated in astrocytes may initially trigger this system, which is eventually eliminated. For the latter, various protein kinases have been proposed as candidates. One of them is mitogen-activated protein kinase (MAPK), which is activated downstream of TLR4 [29,94]. Therefore, the TLR4 ligand stimulation of astrocytes may trigger transcriptional activity via MAPK activation and the phosphorylation of Nrf2.

In addition, inflammatory microglia-derived signaling molecules may activate the Keap1/Nrf2 system in astrocytes. We focused on the NO generated by TLR4 stimulation in microglia cultured with LPS (Figure 3), which not only exerts a cytotoxic role in neurons but also diffuses into astrocytes, where it causes the S-nitrosylation of Keap1, inducing a conformational change and the dissociation of Nrf2 [94]. As a result, the antioxidative roles are triggered in astrocytes. The possibility that neuroprotective astrocytes are induced by such neuroinflammatory microglia was first reported by Shinozaki et al. [104,105]. Inflammatory microglia release several proinflammatory cytokines that induce the downregulation of P2Y1 receptors, which are located upstream of the pathway that induces inflammatory astrocytes, and microglia-derived ATP has been identified as a signaling molecule [104,105]. Although these phenotypic astrocytes can be defined as protective A1 astrocytes, the cytokines they express are actually a mixture of pro-inflammatory and anti-inflammatory factors. Based on the idea that there is no clear distinction between M1 and M2 in microglia or between A1 and A2 in astrocytes, the need for a new definition has been proposed [106].

## 10. Dopamine, Alpha-Synuclein, and TLR4

D2R is known to be minimally expressed in microglia and astrocytes in the resting state [107,108,109]. However, D2R expression is induced in microglia as well as astrocytes in ischemic and inflammatory animal models [107,108,109]. Although the significance of these alterations is not fully understood, D2R may have some neuroprotective effects [94]. For example, D2R agonist stimulation reportedly increases the production of anti-inflammatory cytokines and neurotrophic factors in astrocytes [110]. Furthermore, D2R stimulation reportedly modulates the responsiveness of DAMPs and TLR4 receptors [107,108,109]. In microglia with increased D2R expression, LPS-induced NO production (measured as nitrite production) was evaluated, and NO production was enhanced after D2R agonist administration [107]. As mentioned above, microglia-derived NO may activate the Keap1/Nrf2 system and induce protective astrocytes via the S-nitrosylation of Keap1 in astrocytes [29,94]. Conversely, a reduction in dopamine release, which occurs during the early stage of Parkinson disease, may negatively modulate TLR4-mediated neurotoxic microglia and neuroprotective astrocyte function. The use of D2R agonists for therapeutic intervention in states of receptor hyperexpression is further complicated. Clinical guidelines recommend the use of dopamine agonists in young patients with early Parkinson disease, in preference to l-DOPA. Ropinirole, which is frequently used in clinical practice, is a D2R-selective dopamine agonist, and its glial cell-mediated, disease-modifying effects require further investigation. Particularly, ropinirole itself has a phenolic skeleton and a strong antioxidant activity, so it is expected to exert a protective effect on dopaminergic neurons via its own antioxidant activity [111]. The accumulation of clinical data on ropinirole’s disease-modifying effects is needed.

Alpha-synuclein, a pathological marker of Parkinson disease, is strongly expressed in the nuclei and synapses of neurons and is physiologically assumed to play an important role in synaptic transmission; however, its normal function remains unclear [10,11,95,96]. In the early stage of Parkinson disease, soluble alpha-synuclein fibrils are thought to exert neurotoxic effects, and they may act as a TLR4 ligand to induce neuroinflammation in microglia and astrocytes, leading to further neurological damage [10,11,95,96,112,113]. In animal models, inflammatory changes were observed in astrocytes and microglia in the substantia nigra after the injection of alpha-synuclein into the striatum and prior to pathological changes in the substantia nigra neurons, suggesting that the initial biological response to alpha-synuclein may be TLR4-mediated neuroinflammation [114,115]. In addition, TLR4 stimulation may induce antioxidant stress in astrocytes, triggering neuroprotective effects as described above [29,94]. On the other hand, the aggregates of alpha-synuclein are phagocytosed and removed by astrocytes and microglia. Phagocytosis is particularly prominent in microglia [116]. The TLR4-nuclear factor-κB (NF-κB)-p62 axis is assumed to be the trigger. Few reports, however, have discussed energy metabolism during the phagocytosis of microglia. We will discuss this topic as an unsolved issue in the next chapter.

In addition, we have to recognize that astrocytic or microglial phagocytosis clears alpha-synuclein released from neurons through various pathways but may also spread these toxic molecules to other regions [117,118,119,120,121]. The hypothesis of prion-like propagation of alpha-synuclein has attracted attention of many researchers as an underlying mechanism by which pathological changes and clinical symptoms propagate from the initial site of neuronal degeneration to the whole brain. Especially, microglia move from the original location to the site of inflammation, which may facilitate spreading alpha-synuclein [117,118,119,120,121].

## 11. Issues to Be Resolved in the Future

The compartmentalization of glutamatergic neurons and astrocytes from the perspective of glucose metabolism has been studied in detail [13,29,30,31]. The ANLS model [60,61] is still controversial [62,63,64,65,66,67,68,69,70,71], but adaptations can reasonably explain the activities in some sites of the brain. In contrast, the energy metabolism of microglia is not fully understood. Microglia require physical energy to be transported to inflammatory sites in the brain upon activation. They also require energy to phagocytose damaged and destroyed neurons. NO production from microglia is reportedly dependent on glucose as an energy source [122]. However, the metabolic pathway of glucose as an energy source for microglial function remains largely unknown [123,124,125]. Energy metabolism in inflammatory M1 microglia is believed to shift from oxidative phosphorylation in mitochondria to the glycolytic system [126,127,128], suggesting that PPP flux, a shunt pathway of the glycolytic system, might be enhanced in inflammatory M1 microglia. Just as PPP flux enhancement in astrocytes exerts antioxidant effects on neurons via the antioxidant glutathione, the question of whether such interactions occur between microglia and neurons requires clarification [126,127,128].

Some important findings regarding glutamate metabolism have also been recently reported [129]. Glutamate released from excitatory neurons as a neurotransmitter is synthesized into glutamine via glutamine synthetase in astrocytes and is recycled back to neurons, where it is converted back to glutamate by glutaminase. Ammonia is simultaneously produced; this ammonia is thought to prevent acidification of the intracellular pH and to inhibit cell destruction via lysosomal enzymes in senescent cells. Conversely, an inhibition of glutaminase facilitate removal of senescent cells. However, in the central nervous system, the active removal of senescent cells should not be beneficial for maintaining organ functions, since neuronal cell replacement does not occur easily; therefore, glutaminase inhibition would not be a desirable mechanism for maintaining neuronal function, but the concept might be applicable to microglia [130]. Glutaminase 1 is strongly expressed in microglia in the brain, and the administration of inhibitors to these microglia has been reported to confer a neuroprotective effect [130]. Glutamate metabolism in microglia may be an important target not only for Parkinson disease, but also for other diseases, since microglial activation may play an important role in disease progression.

Intracellular pH reportedly decreases during increased lactate production, but the effect in astrocytes, where this may occur, is unclear. The phagocytosis of astrocytes occurs through autophagy specific for alpha-synuclein, known as synucleinphagy [116,131,132]. This process is reportedly enhanced by a decrease in intracellular pH [131,132]. The enhanced removal of alpha-synuclein through the activation of the glycolytic system can be considered as a series of protective effects that occurs via astrocyte metabolism.

Until now, cultured rodent cells have been used for functional analyses of microglia and astrocytes. Neuroprotective aspects, such as antioxidant effects, have been examined based on the metabolic profile of rodent astrocytes. However, the metabolic profile of human astrocytes is not well known. While researchers have believed the metabolic profile of human astrocytes to be an extension of that in rodents, a recent paper reported that this might not be true [133]. In other words, astrocytes acutely isolated from humans may not necessarily have a high antioxidant activity. On the other hand, we have been inducing astrocytes and spinal motor neurons from human induced pluripotent stem (iPS) cells and analyzing their metabolic profiles [134]. At present, we and other researchers have confirmed that the PPP flux in astrocytes is several times higher than that in neurons and does not differ significantly from the metabolic profile observed in rodent astrocytes [134,135,136]. Further detailed analysis is required. The detailed metabolic profile of human iPS cell-derived microglia, for which differentiation induction methods have recently been established, and their interaction with astrocytes must also be examined.

## 12. Conclusions

The roles of glial cells, especially astrocytes and microglia and their interaction, in pathogenesis and pathophysiology of Parkinson disease were discussed. In Parkinson disease, oxidative stress and mitochondrial dysfunction in hyperactive neurons share a common pathogenetic basis with other neurodegenerative diseases. Importantly, however, dopamine per se induces oxidative stress in contrast to other neurotransmitters. In addition, alpha-synuclein-induced neuroinflammation, which is specific for Parkinson disease, in concert with glial cells propagates pathological processes from the local site to the whole brain. The roles of phagocytosis by astrocytic, as well as microglia in propagation of abnormal protein accumulation, seem to be a double-edged sword. It may reduce toxicity of alpha-synuclein, but also facilitate spreading the toxic protein through travelling of microglia. Nevertheless, astrocytes provide antioxidative function through glucose metabolism, preventing neuronal damage. Microglia may induce an enhancement of astrocytic protective roles via Keap1/Nrf2 system. The interaction of these two glial cells is an attractive research target. Recent findings suggest that astrocytes provide neurons with not only energy substrate like lactate, but also mitochondria. The enhancement of astrocytic mitochondriogenesis and transfer to neurons may serve as therapeutic strategy for neurodegenerative disorders, including Parkinson disease.

## Figures and Tables

**Figure 1 antioxidants-11-00170-f001:**
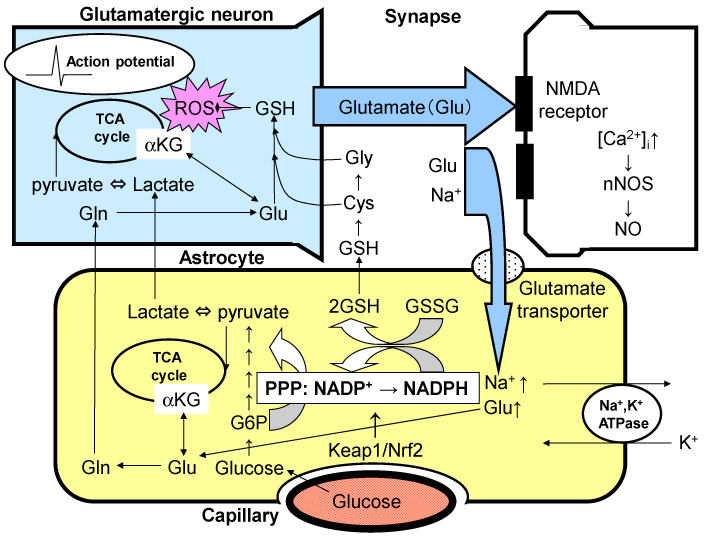
Tripartite synapse consisting of glutamatergic neurons and astrocytic end-foot. (Cited from Takahashi, 2021 [29]).

**Figure 2 antioxidants-11-00170-f002:**
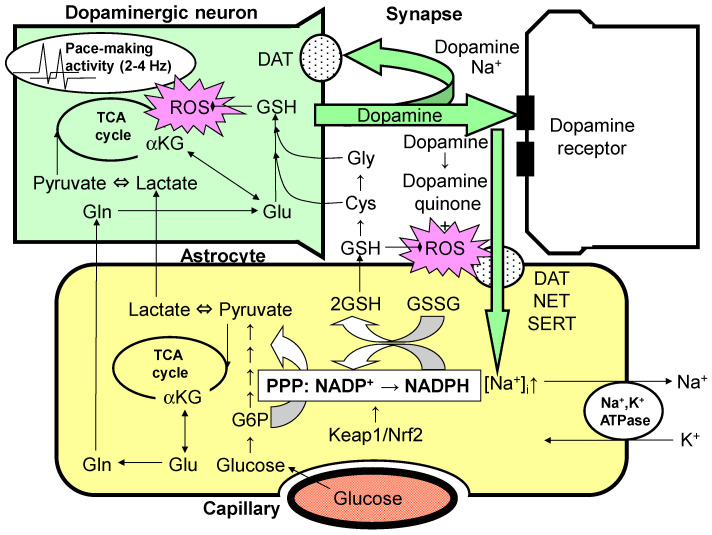
Possible interaction between astrocytes and dopaminergic nerve terminals. (Cited from Takahashi, 2021 [29]).

**Figure 3 antioxidants-11-00170-f003:**
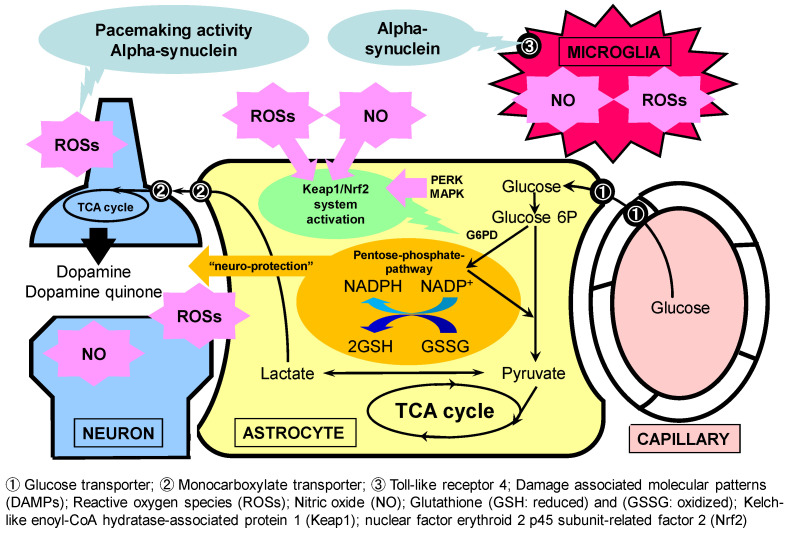
Neuroprotective astrocyte by antioxidants in concert with microglia. Astrocytes are known to play neurotoxic roles in neurodegenerative diseases. Microglia can induce neurotoxic astrocytes through Toll-like receptor (TLR) 4 activation. Reactive oxygen species (ROSs) and nitric oxide (NO) act as neurotoxic molecules that can cause neuronal injury. We focused on the neuroprotective roles of astrocytes through their high glycolysis activity and the pentose–phosphate pathway (PPP) against oxidative stress. Alpha-synuclein and lipopolysaccharide (LPS), a natural TLR4 ligand, induce ROS and NO production in microglia. We found that NO released from microglia activated astroglial PPP flux through the Kelch-like enoyl-CoA hydratase-associated protein 1 (Keap1)/nuclear factor erythroid 2 p45 subunit-related factor 2 (Nrf2) system. Namely, the NO-induced nitrosylation of Keap1 residues released Nrf2 from Keap1, allowing Nrf2 to act as a transcription factor in an in vitro model of cultured rodent microglia and astroglia.

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
