# Peer review of "Neuroprotection and Disease Modification by Astrocytes and Microglia in Parkinson Disease"

_antioxidants, 2022, doi:10.3390/antiox11010170_

Round 1
Reviewer 1 Report
This is outstanding review, perfect elaboration of very complicated molecular mechanisms joining clinical and preclinical data.
I have noticed small editing errors:
p.4 line 177 - "alpha" is lacking before -ketoglutarate ( -KG)
p.10 line 438 - 2x "soluble"
p.5 line 186 - the sentence "GSH can be transfered to neurons via Gln, cysteine (Cys) and glycine (Gly)" seems to me incomprehensible
There is no appeal to figures in the whole manuscript - is it purposeful?
Author Response
- Reviewer #1
This is outstanding review, perfect elaboration of very complicated molecular mechanisms joining clinical and preclinical data. I have noticed small editing errors:
p.4 line 177 - "alpha" is lacking before -ketoglutarate ( -KG)
Ans.1→→→Fig.1 legend (P5,L247): “a” was inserted before “-ketoglutarate” and “KG”
p.10 line 438 - 2x "soluble"
Ans. 2→→→P11, L849: one of two “soluble” s was deleted
p.5 line 186 - the sentence "GSH can be transfered to neurons via Gln, cysteine (Cys) and glycine (Gly)" seems to me incomprehensible
Ans. 3→→→Fig.1 legend (P5,L254-256) "GSH can be transferred to neurons via Gln, cysteine (Cys) and glycine (Gly)" was replaced by "GSH can be transferred to neurons from astrocytes by amino acid components of GSH; i.e., glutathione Gln, cysteine (Cys) and glycine (Gly)."
There is no appeal to figures in the whole manuscript - is it purposeful?
Ans. 4→→→We inserted “(Figure1)”: P3,L156; P4,L216: P4,L223, “(Figure 2)”: P5,L260; P5,L268; P5,L273; P5,L276, and “(Figure 3)”: P9.L726; P9,L734. We think they help the reader to understand better.
Reviewer 2 Report
This is an interesting study, since it addresses the capacity of astrocytes in neuronal protection against neurodegenerative processes such as Parkinson's disease. These protective effects include inflammatory and oxidative processes, normally present in neurodegeneration itself. However, it has important omissions that I would like to highlight:
It only refers to the nigrostriatal dopaminergic pathway without delving into other populations of dopaminergic neurons belonging to the ventral tegmental area of ​​the midbrain which also undergo different degeneration processes in Parkinson's disease. This is the case of the mesolimbic or mesocortical dopaminergic pathway, whose deterioration is key in the pathophysiology of Parkinson's disease, causing emotional and cognitive impairments in these patients. Furthermore, these pathways are mediated by other dopamine receptors such as D3-D4.
Another deficient aspect in the study is the reference to the mechanisms that enable astrocytes in neuronal defense. One of them is the development of astrocytic mitochondriogenesis, which is mediated by mitochondrial transcription factor A (TFAM) and peroxisome proliferator-activated receptor gamma coactivator-1 alpha (PGC-1α), which may inhibit mitochondrial ROS generation and improve mitochondrial respiratory function. In this sense, it would be interesting to assess the role of sirtuin 1 (SIRT-1) and peroxisome proliferator activated receptor γ (PPAR-γ), in the development of astrocyte trainings for neuronal defense.
Author Response
- Reviewer #2
This is an interesting study, since it addresses the capacity of astrocytes in neuronal protection against neurodegenerative processes such as Parkinson's disease. These protective effects include inflammatory and oxidative processes, normally present in neurodegeneration itself. However, it has important omissions that I would like to highlight:
It only refers to the nigrostriatal dopaminergic pathway without delving into other populations of dopaminergic neurons belonging to the ventral tegmental area of the midbrain which also undergo different degeneration processes in Parkinson's disease. This is the case of the mesolimbic or mesocortical dopaminergic pathway, whose deterioration is key in the pathophysiology of Parkinson's disease, causing emotional and cognitive impairments in these patients. Furthermore, these pathways are mediated by other dopamine receptors such as D3-D4.
Ans. 1
→→→In “1. Introduction”, we added some explanation about the VTA, which exhibits miler and slower degeneration as compared with SN with appropriate refs: P1,L34-38.
→→→In “1. Introduction”, We added one possible explanation for miler and slower degeneration of dopaminergic neurons in VTA: Ca2+-induced oxidative stress: P2,L63-66.
→→→In “2. Dopaminergic neurons and astrocytes”, We mentioned dopamine D3,4 receptors that are expressed in the mesolimbic and mesocortical systems. Unfortunately, however, we could not find the mechanistic explanation by which these receptors affect the neurodegenerative processes in the VTA. We described as it is: P2,L95-98.
Another deficient aspect in the study is the reference to the mechanisms that enable astrocytes in neuronal defense. One of them is the development of astrocytic mitochondriogenesis, which is mediated by mitochondrial transcription factor A (TFAM) and peroxisome proliferator-activated receptor gamma coactivator-1 alpha (PGC-1α), which may inhibit mitochondrial ROS generation and improve mitochondrial respiratory function. In this sense, it would be interesting to assess the role of sirtuin 1 (SIRT-1) and peroxisome proliferator activated receptor γ (PPAR-γ), in the development of astrocyte trainings for neuronal defense.
Ans. 2→→→In “4. Dopamine and astrocyte activation of the glycolytic system and PPP ”, we added the recent hypothesis regarding astrocytic mitochondrial transfer to neurons. The dysfunction and degeneration of mitochondria in dopaminergic neurons play crucial roles in the pathogenesis of PD (P5,L277-288). Activation of mitochondriogenesis in not only in neurons but also in astrocytes could slow the disease progression of PD (P5, L289-P6, L392). This is now a partially realistic idea in stroke therapy. We appreciate the reviewer’s important suggestion.
Reviewer 3 Report
This draft summarizes the role of astrocytes and microglia in PD. This review paper should be of interest to the related research field and is highly acceptable for antioxidants but must address some major issues.
- Prion-like spreading of alpha-synuclein is the major contributing factor for PD pathogenesis. The role of glial cells in this spreading should be included.
- Pagee5-6, line 208-236: There is no reference in this paragraph.
- Page10, line 399-411: There is no reference in this paragraph.
- Section 5 (PPP activation~): PPP activation in PD is not explained in this section.
- Pathologic alterations of astrocytes in PD and their role in neurodegeneration are well-known. The authors should be discussed glial cell-mediated neurodegeneration in PD.
Author Response
- Reviewer #3
This draft summarizes the role of astrocytes and microglia in PD. This review paper should be of interest to the related research field and is highly acceptable for antioxidants but must address some major issues.
- Prion-like spreading of alpha-synuclein is the major contributing factor for PD pathogenesis. The role of glial cells in this spreading should be included.
Ans. 1→→→We have added the discussion as the 3rd para of “10. Dopamine, alpha-synuclein, and TLR4”: P11,L863-870.
- Pagee5-6, line 208-236: There is no reference in this paragraph.
Ans. 2→→→As answered in the beginning (Response to Editorial Manager (and Reviewer #3)), this part is not body text, but figure legend. We reformatted it to make this point clearer: P6,L408-P7, L442.
- Page10, line 399-411: There is no reference in this paragraph.
Ans. 3→→→As answered in the beginning (Response to Editorial Manager (and Reviewer #3)), this part is not body text, but figure legend. We reformatted it to make this point clearer: P10,L774-785.
- Section 5 (PPP activation~): PPP activation in PD is not explained in this section.
Ans. 4→→→We are sorry about our careless mistake. As the reviewer has pointed out, this “” section is meaningless. The text in this section was moved to the end of the previous section (P6, L393-405). As a result, “5. PPP activation in astrocytes and Parkinson disease” was deleted and “4. Dopamine and astrocyte activation of the glycolytic system” has been renamed as “4. Dopamine and astrocyte activation of the glycolytic system and PPP”: P5,L258
- Pathologic alterations of astrocytes in PD and their role in neurodegeneration are well-known. The authors should be discussed glial cell-mediated neurodegeneration in PD.
Ans, 5→→→As the reviewer has pointed out in the very first comment. Prion-like spreading of alpha-synuclein is an important aspect for this review paper. We have added the discussion in the 3rd para of “10. Dopamine, alpha-synuclein, and TLR4” (P11,L863-870) and “12. Conclusions” (P12,L950-967).
Round 2
Reviewer 2 Report
The changes made provide more appropriate content.Altogether it is a study of remarkable quality
Reviewer 3 Report
All reviewer's concerns were completely addressed in the revised manuscript. Therefore, the current version of the manuscript is now suitable for publication in antioxidants.